# Secondary Metabolites and Antiradical Activity of Liquid Fermentation of *Morchella* sp. Isolated from Southwest China

**DOI:** 10.3390/molecules24091706

**Published:** 2019-05-02

**Authors:** Cailing Yang, Xuan Zhou, Qingfeng Meng, Mengjiao Wang, Yao Zhang, Shaobin Fu

**Affiliations:** 1Pharmacy School of Zunyi Medical University, Zunyi 563000, China; ycailing@126.com (C.Y.); 15764535094@163.com (X.Z.); wmj1585506623@163.com (M.W.); 13087864173@163.com (Y.Z.); 2Department of Public Health, Zunyi Medical University, Zunyi 563000, China; qfmeng@126.com

**Keywords:** *Morchella*, strain identification, liquid fermentation, chemical constituents, antiradical activity

## Abstract

Morels famous for their taste and nutrition are in short supply all over the world although they were considered as one of the most highly prized edible and medicinal mushrooms. Because of the limitation of resource and cultivation technology, fermentation of edible mushroom was gradually applied to nutrient, bioactivity and breeder seed preparation. At present, there are more reports on sugar and amino acid but less on other components. *Morchella* sp. YDJ-ZY-1 was isolated from the wild fruiting body by the spores releasing method in Zunyi Guizhou province in Southwest China and identified based on phenotype and genotype characteristics. Chemical compositions of YDJ-ZY-1 were investigated from liquid fermentation that will lay the foundation for further development and utilization. Four pyranoids (1–4) and 2-(1-oxo-2-hydroxyethyl) furan (5), linoleic acid (6), Morelin (2-hydroxy-cinnamic acid methyl ester, (7) and 1-*O*-β-d-ribofuranose-Morelin (8) were obtained from EtOAc extraction and elucidated by spectral data. Product 4 and 8 were new compounds and 7 was isolated from nature for the first time. Antiradical activity was evaluated by free radical scavenging effect on DPPH (1,1-Diphenyl-2-picrylhydrazyl radical 2,2-Diphenyl-1-(2,4,6-trinitrophenyl)hydrazyl). Compound **5** exhibited strong antiradical activity while compounds **1** and **2** exhibited moderate activity. Thus, incubation of *Morchella* sp YDJ-ZY-1 separated from the wild fruit body afforded eight compounds. Secondary metabolites with new structures were mined from fermentation of *Morchella* sp. and antiradical activity was evaluated.

## 1. Introduction

Morels are some of the most highly prized edible and medicinal mushrooms, and outdoor cultivation has been achieved in recent years. These edible mushrooms appreciated worldwide have been used in traditional medicine for centuries, due to their health-related benefits. In recent research, the anti-oxidative, anti-inflammatory bioactivities, immunostimulatory, anti-tumor and antimicrobial properties were reported [1,2,3]. It was considered that morel’s health benefits were attributed mainly to sugar (polysaccharides) and to various constituents such as compounds, amino acids, important vitamin, fatty, organic acid and mineral profile [2,4]. The crude polysaccharide isolated from the fruiting bodies of *Morchella importuna* showed a neuroprotective effect against H_2_O_2_-induced PC12 cell cytotoxicity by reducing oxidative stress [5]. Due to unique flavor, taste and texture, morels are used in different recipes all over the world. Additionally, morels are used as a laxative, purgative, emollient, body tonic, and for stomach problems, healing wound and for general weakness [6]. Benefited from its high price, morels play a very important role in the economy of the country. Due to the high demand for morels and their increasing economic importance, research on its cultivation, bioactivity, functional compositions is being investigated more and more [6]. Anti-tumor bioactivity-guided isolation of the MeOH extraction of *Morchella esculenta* afforded eight compounds including three fatty acids and five sterols. Three compounds exhibited the most potent cytotoxicity to human lung cancer cell lines [7]. *Morchella* protein hydrolyzate (MPH) produced by *M. esculenta* exhibits antioxidant activity [8]. The carboxymethylated polysaccharide (CPMEP) derived from the purified polysaccharide from *Morchella angusticepes* Peck (PMEP) showed stronger cholesterol-lowering activity than PMEP in rats [9]. Because of the limitation of wild *Morchella* resources, the cultivation of *Morchella* is increasingly popular for consumption as a functional food and for food-flavoring. Due to the limitation of strict techniques and difficulty in large-scale cultivation, fermentation has attracted more and more attention. It is of significance and potential to develop approaches to replace the traditional supply chain. However, there is no systematic research on the chemistry of liquid fermentation of *Morchella*. 

Herein, a strain, YDJ-ZY-1, was isolated from wild *Morchella* in Zunyi Guizhou province and fermented in PDA media. Eight compounds were obtained from EtOAc extraction. Products 4 and 8 were new compounds and 7 was a new natural product. The antiradical activity was evaluated. 

## 2. Results

### 2.1. Identification of Strain 

The strain YDJ-ZY-1 was obtained from wild *Morchella* by the spore releasing method. Hyphae were thick, strong, white mycelium at the beginning while yellowish in the middle and later stage, then the color changed to deeper and brownish in the later stage (Figure 1). Brownish sclerotia with different sizes were easily produced. The capsule is cylindrical. Eight ascospores arranged in a single row are elliptical and colorless (400×) (Figure 2). Genomic DNA of the strain was extracted by CTAB (cetyltriethylammnonium bromide) method and identified as *Morchella* sp. with 99% similarity based on internal transcribed spacer (ITS) of ribosomal DNA (rDNA). Phylogenetic tree by the Neibor-joining method was constructed by Mega 5.1 (Figure 3). 

### 2.2. Isolation and Identification of Secondary Metabolites 

Eight compounds were isolated and purified by repeated column chromatography with different eluting agents from fermentation (Figure 4). The antiradical activity was evaluated by the DPPH method. 

Compound **1** was obtained as brown oil. The ^13^C-NMR spectrum (183.1, 171.6, 169.2, 114.3, 111.5) revealed the parent skeleton of pyrone type of which indicated two substituted carbons signals (171.6, 169.2). There were also oxygen-bearing methylene carbon signals at δ 61.1, as well as a methyl carbon at δ 19.6. Two aromatic ring protons signals 6.18 (s), 6.36 (s) provided evidence that 1 was structure with two substituted pyrone consistent with ^13^C-NMR. Two methoxy protons signals at δ 4.41 (s, 2H) and three methyl protons signals at δ 2.32 (s, 3H) were observed in ^1^H-NMR. From HMBC (Heteronuclear Multiple-Bond Correlation) data, methoxy protons signals at δ 4.41 were correlated with carbons signals δ_C_ 111.5 and δ_C_ 171.6. Methyl protons signals at δ 2.32 were related with δ_C_ 114.3 and δ_C_ 169.2. Aromatic ring protons at δ_H_ 6.18 was related with δ_C_ 19.6, δ_C_ 111.5 and δ_C_ 169.2 while δ_H_ 6.36 was related with δ_C_ 114.3, δ_C_ 171.6 and δ_C_ 61.1 in HMBC (Figure 5). Compared ^1^H-NMR and ^13^C-NMR spectral data with the literature [10,11], its structure was confirmed as 1.

Compound **2** was isolated as brown oil. In the ^1^H-NMR spectrum, one olefinic protons signals at δ 5.22 (s, 1H), two oxygen-bearing protons signals at δ 4.44 (m, 1H) and 3.75 (t, 1H), three methyl protons signals at δ 1.98 (s, 3H) were observed. The ^13^C-NMR spectrum of 2 showed one carbonyl signal at δc 192.1. Corresponding to the ^1^H-NMR spectrum, there were two oxygen-bearing carbon signals δc 80.9 and δc 65.6 in ^13^C-NMR. Compared NMR data with those reported in the literatures [12], the structure of 2 was determined to be Erinapyrone B. 

Compound **3** was obtained as a white solid. The molecular formula was determined to be C_8_H_10_O_4_ on the basis of HR-ESI–MS data M + Na]^+^ ion at *m*/*z* 193.0466 (calcd 192.0477), [2M + Na]^+^ ion at *m*/*z* 363.1001 (calcd 363.1056). Similar to compound **1**, compound **3** also showed pyrone skeleton with carbon signals δc 181.3, δc 170.5, δc 167.8, δc 124.7, δc 111.6. However, there was one more oxygen-bearing carbon signal δc 54.9 directly connected with δ_H_ 4.51 (s, 2H) except for δc 61.1 at C7 that directly related to δ_H_ 4.43 (s, 2H) in HSQC. Besides, one aromatic ring proton disappeared. Thus, hydroxymethylation reaction maybe occurred for compound **3** from **1**. The signal of δ_H_ 4.51 (s, 2H) correlated with δ_C_ 124.7 (C3), δ_C_ 167.8 (C2), δ_C_ 181.3 (C4) in HMBC spectrum (Figure 5) indicated the hydroxymethylation maybe at position C3. Data of NMR and MS of compound **3** were consistent with literature that was also isolated from edible mushrooms [13]. Therefore, compound **3** was confirmed as 3,6-bis(hydroxymethyl)-2-methyl-4H-pyran-4-one, herierin III. 

Compound **4** was isolated as brown oil. It displayed [M + Na]^+^
*m*/*z* at 235.0601 (calcd 235.0582), [2M + Na]^+^
*m*/*z* at 447.1215 (calcd 447.1267) on HR-ESI-MS, corresponding to the molecular formula of C_10_H_12_O_5_. Based on the ^1^H-NMR and ^13^C-NMR, compound **4** showed the similar parent structure with compound **3**. Five carbon signals δc 180.9, δc 170.9, δc 169.2, δc 120.8, δc 2111.5 and one aromatic ring proton signal δ_H_ 6.39 (s, 1H) indicated the three substituted pyrone. Two oxygen-bearing carbons signals at δ_C_ 61.0 related to δ_H_ 4.40 (s, 2H) and δ_C_ 57.8 related to δ_H_ 4.97 (s, 2H) in HMBC spectrum were observed. Long-range correlation between δ_H_ 4.40 with C4 (δ_C_ 180.9), C5 (δ_C_ 111.5) and C6 (δ_C_ 170.9) in HMBC showed the hydroxymethyl group was at C6 that is the same in compound **3**. However, there was one more carbonyl carbon signal (δ_C_ 172.8) and methyl carbon signal (δ_C_ 20.7) except for C8 (δ_C_ 17.7). δ_H_ 2.01 (s, 3H) was directly connected with δ_C_ 20.7 in HSQC and had strong long correlation with carbonyl carbon C10 (δ_C_ 172.8) and weakly long correlation with C9 (δ_C_ 57.8) in HMBC (Figure 5). On the basis of above evidences, product 4 was elucidated as (6-(hydroxymethyl)-2-methyl-4-oxo-4H-pyran-3-yl) methyl acetate (herierin V) that was a new compound. 

Compound **5** was obtained as brown amorphous solid. There were four olefinic carbons signals at δ 150.2, 118.0, 112.6 and 147.2, one carbonyl carbon signal at δ 187.8 and one oxygen-bearing methylene carbon signal at δ 65.2 in the ^13^C-NMR spectrum. The ^1^H-NMR spectrum showed three olefinic protons signals at δ 7.59 (s, 1H), 7.26 (d, *J* = 3.6 Hz, 1H), and 6.55 (m, 1H). Corresponding to carbon spectrum, two oxygen-bearing protons signals at δ 4.69 (s, 2H) were observed. Based on these data and comparison with literature, compound **7** was identified as 2-(1-oxo-2-hydroxyethyl) furan [14] (Figure 4).

Compound **6** was isolated as white powder. ^13^C-NMR spectra of 6 were typical of a fatty acid. The ^13^C-NMR spectrum revealed four olefinic carbons signals at δ 130.3, 130.1, 128.2 and 128.0, as well as a carboxyl carbon signal at δ 180.6. Additionally, there were serials of methylene carbons signals at δ 24.8–34.3. The ^1^H-NMR spectrum exhibited the presence of olefinic protons signals at δ 5.37 (m) and methyl group 0.88 (m). By a comparison with data available in the literature, compound **6** was determined to be linoleic acid [15].

Compounds **7** and **8** were isolated as white amorphous powder. Compared the ^13^C-NMR and ^1^H-NMR, compounds **7** and **8** showed the similar structures. Compound **8** maybe the glycoside product of 7. ^13^C-NMR spectroscopic data indicated compound **8** was a derivative of substituted aromatics ring with carbon signals at 124.6 (C1), 155.6 (C2), 116.0 (C3), 131.8 (C4), 122.4 (C5), 127.7 (C6). Besides, there was two olefinic carbon signals at δ 140.2 (C7) and 121.1 (C8), an ester carbonyl signal at 168.6 (C9), as well as pentose signals at 101.0, 73.0, 70.9, 86.4, 62.9. In ^1^H-NMR, four aromatics ring proton signals at 7.20 (d, *J* = 8.3 Hz, 1H), 7.31 (t, *J* = 7.7 Hz, 1H), 7.00 (t, *J* = 7.5 Hz, 1H), 7.50 (d, *J* = 7.7 Hz, 1H) and two olefinic proton signals at 8.01 (d, *J* = 16.2 Hz, 1H), 6.40 (d, *J* = 16.2 Hz, 1H) that indicated a two substituted benzene ring and a double bond of 8 with trans-configuration. In HSQC spectrum, correlation between 116.0 (C3) with δ_H_ 7.20 (H-2), 131.8 (C4) with δ_H_ 7.31 (H4), 122.4 (C5) with δ_H_ 7.00 (H5), 127.7 (C6) with δ_H_ 7.50 (H6) as well as, δ_C_ 140.5 with δ_H_ 8.01, δ_C_ 117.5 with δ_H_ 6.40. From HMBC spectra (Figure 5), H1′ (d, *J* = 3.3 Hz, 1H) was related to 155.6 (C2), H8 (6.40, d, *J* = 16.2 Hz, 1H) to C1 (124.6), C7 (140.5), C9 (168.6), H7 (8.01, d, *J* = 16.2 Hz, 1H) to C8 (117.5), C6 (127.7), C2 (155.6), C9 (168.6) that suggesting the sugar was connected to C2 and another side chain was signed to C1. Furthermore, oxygen-bearing methyl protons signals 3.47 (s, 3H) showed long range correlation with C9. The pentose was identified as β-d-ribofuranose according to chemical shift of carbons in Table 1. Oxygen-bearing methylene were observed at δ_H_ 3.76 (2H), δ_C_ 62.9 in compound **8,** suggesting the ribofuranose. The chemical shifts were consistent with β-d-ribofuranose. H1′ was related to 155.6 (C1) revealed the sugar was connected to C1. The HR-ESI-MS indicated the molecular formula C_15_H_1__8_O_7_ by [M + Na]^+^ 333.1128 (calcd 333.0950), [2M + Na]^+^ 643.2215 (calcd 643.2003). Compound **7** showed the similar carbon and proton signals except sugar. There were two substituted 158.3, 142.4, 132.7, 130.0, 122.5, 120.8, 117.0, one carbonyl group 170.0. Compound **7** was the aglycone of 8. It was isolated from natural for the first time, so it was named as Morelin. Compound **8** named as 1-*O*-β-d-ribofuranose-Morelin that was a new product. 

### 2.3. Antiradical Activity of Compounds 

Ability of scavenging effect on DPPH free radical was accomplished to evaluate the antioxidant activity. Five concentrations were tested in the experiment, Vitamin C (VC) was the positive control (Figure 6). The data obtained in the present study suggests that compound **5** showed strong scavenging effect on DPPH free radical activity while compounds **1** and **2** exhibited moderate activity against free radicals. Compounds **3**, **4**, **7** and **8** almost exhibited no effect. For compounds **1** and **2**, the radical scavenging capacity increased as the increasing of compound concentration. Compared to **1** and **2**, compounds **3** and **4** exhibited less activity while the difference was the one more substitute side chain at C3 in compounds **3** and **4**. The antiradical activity may be affected by the side chain. Moreover, the length of the side chain has little effect on the activity. Statistical analysis indicated that clearance of different compounds at the same concentration is different. The clearance of the same compound is different at different concentrations. Compound **5** showed the best clearance at 0.25 mg/mL.

## 3. Discussion

*Morchella* are famous rare edible fungi with high medicinal and economic value that show great potential in food, health products, pharmaceutical and cosmetics. The effects of *Morchella* on immune regulation, preventing decrepitude, anti-tumor, anti-fatigue, protection of the cardiovascular system and liver are proven in pharmacological tests. With the development of modern techniques, new effective metabolites and active monomers are eager to be discovered. Polysaccharide of *Morchella*, the main active ingredient, was mostly investigated. Additionally, there are many other kinds of chemical constituents (amino acids, sterols, organic acids et al.) found from the fruiting body of the *Morchella* species. However, there are no reports on the chemistry of liquid fermentation. Strain *Morchella* sp.YDJ-ZY-1 was isolated from wild fruiting body in Zunyi Guizhou province, Southwest China. Chemical compositions of EtOAc extraction from liquid fermentation were investigated and eight secondary metabolites were obtained. Compound **1**–**4** were structures of pyrone, compound **5** was furan derivative, compound **6** was unsaturated fatty acid, and compounds **7** and **8** were methyl cinnamate derivatives. Product 4 and 8 were new compounds while product 7 was a new natural product isolated from nature for the first time. Metabolite 5 with strong radical scavenging capacity was found the highest yield (3.4%). Product 6 linoleic acid, one of essential fatty acids that must be obtained through diet was proved important in metabolism and health. It could effectively reverse the inflammatory responses induced by palmitic acid treatment in microglial cells [16]. Research points to linoleic acid’s anti-inflammatory, acne reductive, skin-lightening and moisture retentive properties when applied topically on the skin [17]. Product 7 and 8 with the structure of methyl cinnamate is mainly used in the daily chemical and food industry. Methyl cinnamate a flavor agent, is commonly used as fixative or edible flavor, but is also an important organic synthetic raw material. Methyl cinnamate is used as a chemical identification and distinguishing *Tricholoma matsutake* and *Agaricus blazei* and it could also provide a reference for evaluating the quality of edible fungi *Tricholoma matsutake* and *Agaricus blazei* [18]. Because of it is rich in bioactive ingredients and has many pharmacological effects, *Morchella* has broad prospects for development and utilization. 

## 4. Materials and Methods 

### 4.1. Isolation and Identification of Morchella sp. YDJ-ZY-1 

Species of wild *Morchella* were isolated by spore releasing method by a self-made isolation device [19]. The sprayed spores were collected at different times by agar water medium, placed directly below and around the isolation device. When the spores were germinated, hyphae were transformed and purified into PDA (Potato Dextrose Agar) slant and then stored in 4 °C. The genomic DNA was extracted and ITS (internal transcribed spacer) sequence was applied to identify strain combined microstructure. The primers ITS1F (5′-CTTG GTCA TTTA GACG AAGTAA′) and ITS 4(5′- TCCT CCGC TTAT TGAT ATGC-3′) were used to amplify the ITS rDNA. The PCR (Polymerase Chain Reaction) reaction was performed with the following cycles: (1) 94 °C for 3 min; (2) 30 cycles of 94 °C for 1 min, 55 °C for 1 min and 72 °C for 2 min, and (3) 72 °C for 5 min. Sequences were accomplished by Beijing Boyoushun Bio-ulab Technology Co., LTD. Homologous analysis was carried out by BLAST in NCBI website (https://blast.ncbi.nlm.nih.gov/Blast.cgi). Phylogenetic tree was constructed based on the Neighbor-joining method by Mega 5.1. 

### 4.2. Fermentation of Strain

The purified cultures were incubated in 500 mL Erlenmeyer flasks that contain 200 mL PDA liquid media at 18–20 °C and 135 rpm on a shaker. And then cultures were transformed into twenty 2000 mL flasks containing 1000 mL PDA media for another 10 days. 

### 4.3. Isolation and Identification of Metabolites 

After fermentation, the mycelia and broth were separated by filtration. Then they were extracted three times with EtOAc. The extracts were evaporated under reduced pressure to afford 6.88 g crude extract. The crude residue was separated on medium pressure C18 column chromatography with MeOH-H_2_O (10% to 100%, *v*/*v*) to afford nine fractions (Fraction 1–9). Fraction 2 was subjected to column chromatography on silica gel (300–400 mesh) eluted with a stepwise petroleum ether/ ethyl acetate (1:1 to 0:1, *v*/*v*) to yield compound **2** (14 mg) and compound **3** (40 mg). Compound **1** (18 mg) was prepared by silica gel column chromatography gradient eluted with petroleum ether: ethyl acetate (3:1 to 1:1, *v*/*v*). Fraction 5 was eluted with a stepwise petroleum ether/ethyl acetate (7:1) to afford compound **5** (236 mg). Compound **4** (3.0 mg) was prepared by silica gel column chromatography with dichloromethane: methanol = 25:1 (*v*/*v*) with equivalence elution and preparation thin liquid chromatography (*p*-TLC) (ethyl acetate: tetrahydrofuran=1.5:1, *v*/*v*). Compound **8** (9.3 mg) was obtained by silica gel column chromatography eluted with ethyl acetate from Fraction 7 while compound **7** (14.5 mg) was purified by petroleum ether/ethyl acetate (2:1) from Fraction 8. Compound **6** (16 mg) was afforded by petroleum ether/dichloromethane (8:1) and petroleum ether/ ethyl acetate (8:1) from Fraction 9.

Structures of products were elucidated by NMR, MS and references. NMR spectra were acquired on a Agilent DD2400-MR spectrometer operating at 400 MHz (for proton NMR) and 100 MHz (for carbon NMR). High resolution mass spectra were obtained in ABSciex TOF500^+^ MS using electrospray ionization. 

Compound **1**: Brown oil, ^1^H-NMR (400 MHz, MeOD): 6.18 (s, H-5), 6.36 (s, H-3), 4.41 (s, 2H, H-7), 19.6 (s, 3H, H-8). ^13^C-NMR (100 MHz, MeOD): 183.1(C4), 171.4(C6), 169.2(C2), 114.3(C3), 111.5(C5), 61.1(C7), 19.6(C8).

Compound **2**: Brown oil, ^1^H-NMR (400 MHz, MeOD): 5.22(s, H-3), 4.44 (m, H-6), 3.82 (m, H-7), 3.75 (m, H-7), 2.54 (m, H-5), 2.25 (m, H-5), 1.98 (s, 3H, H-8). ^13^C-NMR (100 MHz, MeOD): 192.1 (C4), 174.1 (C2), 105.0 (C3), 80.9 (C6), 63.9 (C8), 37.7 (C5), 20.8 (C7). 

Compound **3**: White solid, HR-ESI-MS: 193.0466[M + Na]^+^, 363.1001[2M + Na]^+^. ^1^ H-NMR (400 MHz, MeOD): 6.4 (s, H-5), 4.51 (s, 2H, H-9), 2.46 (s, 3H, H-8). ^13^C NMR (100 MHz, MeOD): 181.3 (C4), 170.5 (C6), 167.8 (C2), 124.7 (C3), 111.6 (C5), 54.9 (C9), 37.7 (C5), 20.8 (C7). 

Compound **4**: Brown oil, HR-ESI-MS: 235.0601 [M + Na]^+^, 447.1215[2M + Na]^+^. ^1^ H-NMR (400 MHz, MeOD): 6.39 (s, H-5), 4.97 (s, 2H, H-9), 4.40 (s, 2H, H-7), 2.44 (s, 3H, H-8), 2.01 (s, 3H, H-11). ^13^C-NMR (100 MHz, MeOD): 180.9 (C4), 172.8 (C10), 170.9 (C6), 169.2 (C2), 120.8 (C3), 111.5 (C5), 57.8 (C9), 20.7 (C11), 17.7 (C8). 

Compound **5**: Brown solid, ^1^ H-NMR (100 MHz, CDCl_3_): 7.59 (s, 1H, H-5), 7.26 (d, *J*= 3.6 Hz, 1H, H-3), 6.55 (m, 1H, H-4), 4.69 (s, 2H, H-2′); ^13^C-NMR (100 MHz, CDCl_3_): 187.7 (C-1′), 150.0 (C-2), 147.2 (C-5), 118.0 (C-3), 112.6 (C-4), 65.2 (C-2′).

Compound **6**: White solid, ^1^H-NMR (400 MHz, CDCl_3_): 5.37 (m, H9,10,12,13), 0.88 (m, H-18). ^13^C-NMR (100 MHz, ClCD_3_): 180.5 (–COOH), 130.3 (C9), 130.1 (C12), 128.2 (C10), 128.0 (C11), 34.3~24.8 (–CH2), 14.3 (3H, H-18).

Compound **7**: White amorphous power, ^1^H-NMR (400 MHz, MeOD): 7.99 (d, *J*= 16.0 Hz, H-7), 6.61(d, *J*= 16.0 Hz, H-8), 7.49 (d, *J*= 8.0 Hz, H-3), 7.22 (t, *J*= 8.0, 16.0 Hz, H-6), 6.84–6.86 (m, H-4, 5), 3.79 (s, H-10). ^13^C-NMR (100 MHz, MeOD): 170.0 (C9), 158.3 (C2), 142.4 (C7), 132.7 (C3), 130.0 (C4), 122.5 (C5), 117.0 (C8), 52.0 (C10).

Compound **8**: White amorphous power, HR-ESI-MS: 333.1128[M + Na]^+^, 643.2215 [2M + Na]^+^. ^1^H-NMR (400 MHz, MeOD): 8.01 (d, *J*= 16.2 Hz, H-7), 7.50 (d, *J*= 7.7 Hz, H-6), 7.31 (d, *J*= 7.7 Hz, H-4), 7.20 (d, *J*= 8.3 Hz, H-3), 7.00 (t, *J*= 7.5 Hz, H-5), 6.40 (d, *J*= 16.2 Hz, H-8), 5.69 (d, *J*= 3.3 Hz, H-1′), 4.37 (s, H-4′), 4.26 (o, 2H, H2′-3′), 3.76 (t, 2H, H-5′). ^13^C-NMR (400 MHz, MeOD): 168.6 (C9), 155.6 (C2), 140.5 (C7), 131.8 (C4), 127.7 (C6), 122.4 (C5), 117.5 (C8), 116.0 (C3), 101.0 (C1′), 86.4 (C4′), 73.0 (C2′), 70.9 (C3′), 62.9 (C5′), 51.7 (C10). 

### 4.4. Antiradical Activity of Compounds 

In this study, DPPH radical scavenging activity was analyzed for preliminary evaluation on the antioxidant activity of purified products. The radical scavenging activity of purified products was determined with slight modifications in the method [20]. A total of 2.0 mL from a 0.1 mM ethanol solution of the DPPH radical was mixed to 1.0  mL sample. Series concentrations of compounds **1**, **3**, **5** and positive control drug Vitamin C (VC) were 2 mg/mL, 1 mg/mL, 0.5 mg/mL, 0.25 mg/mL, 0.125 mg/mL, respectively, while concentrations of compounds **2**, **4**, **7** and **8** were 0.80 mg/mL, 0.40 mg/mL, 0.20 mg/mL, 0.10 mg/mL, 0.05 mg/mL. The mixtures were well shaken and kept at room temperature in the dark for 10 min. The absorbance was measured at 516 nm using a double beam UV-VIS spectrophotometer. Methanol was used as a negative control and VC was positive control. All the tests and the controls were repeated in triplicate. The free radical scavenging effect was evaluated by K(%) = [(Ao − As)/Ao]*100, Ao: 2 mL DPPH + 1 mL methanol, As: 2 mL DPPH + 1 mL sample. 

## 5. Conclusions

Fermentation of *Morchella* sp YDJ-ZY-1 was accomplished to accumulate biomass and incubation of *Morchella* sp afforded eight compounds including two new products and one new natural compound. Secondary metabolites with new structures were mined from wild type strain. Antiradical activity of *Morchella* sp. was evaluated by DPPH method and compound **5** showed strong activity.

## 6. Patents

A self-made isolation device of *Morchella* sp. China (ZL 201620445823.X).

## Figures and Tables

**Figure 1 molecules-24-01706-f001:**
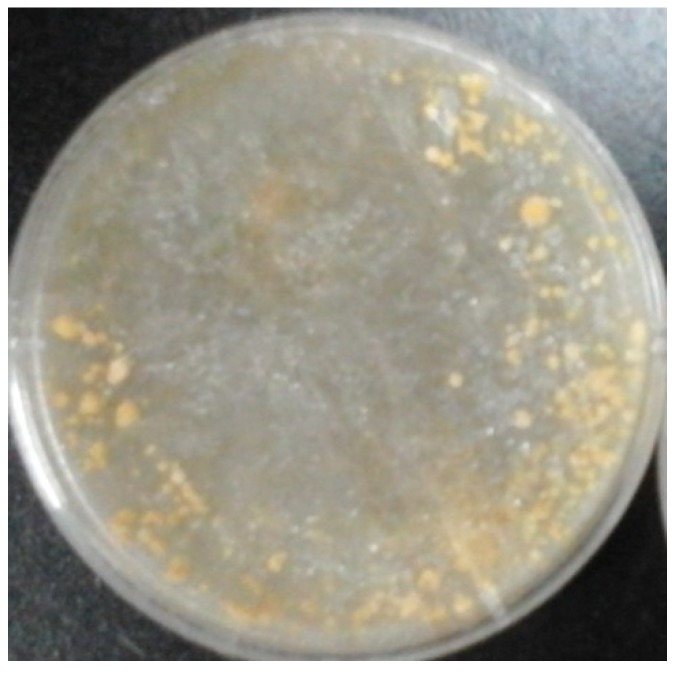
Colony of YDJ-ZY-1.

**Figure 2 molecules-24-01706-f002:**
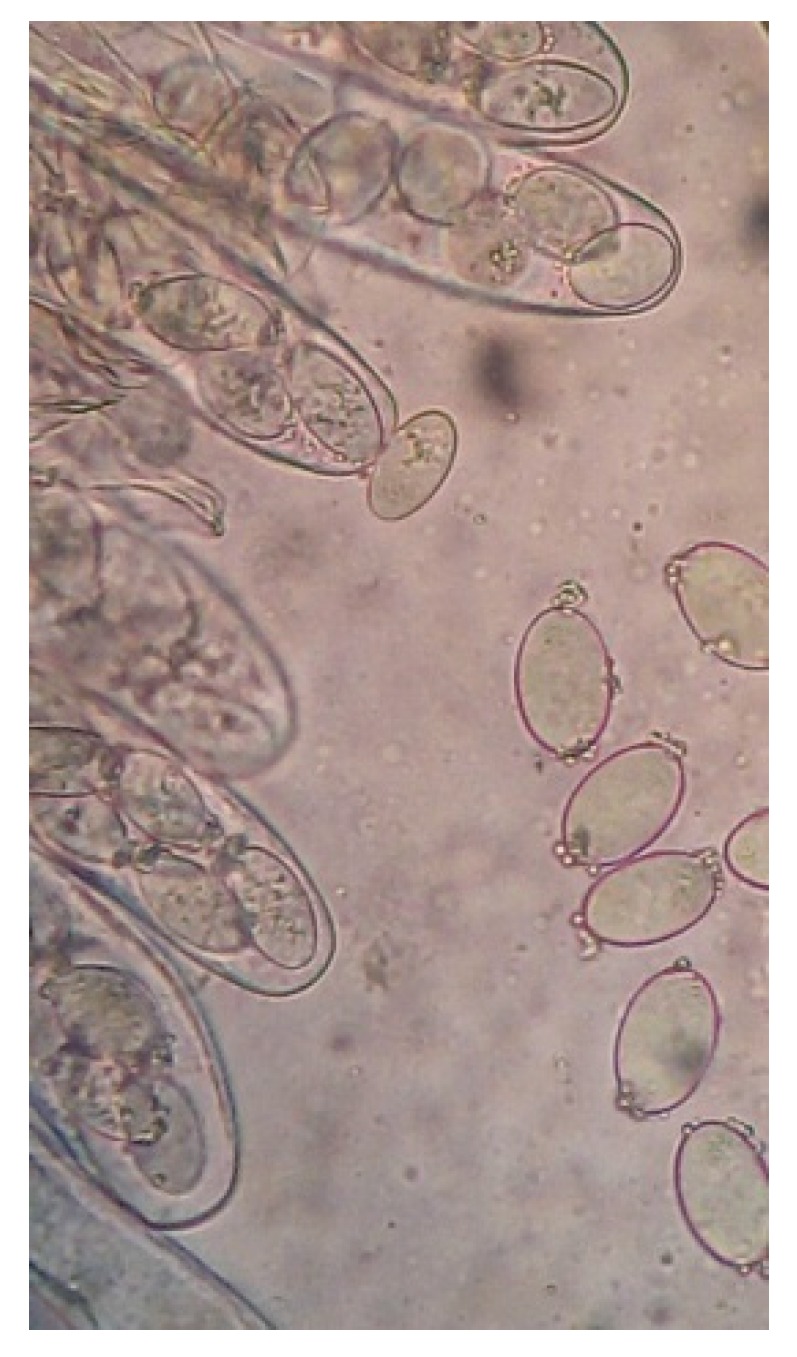
Spores of YDJ-ZY-1.

**Figure 3 molecules-24-01706-f003:**
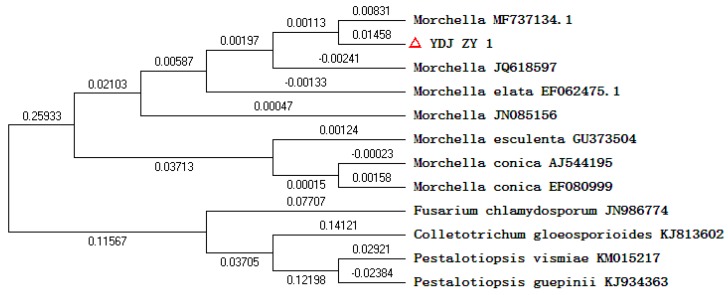
Phylogenetic tree by Neibor-joining method was constructed by Mega 5.1.

**Figure 4 molecules-24-01706-f004:**
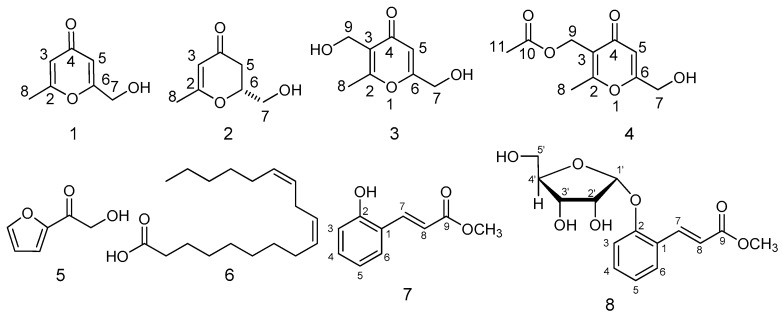
Structures of compounds **1**–**8**.

**Figure 5 molecules-24-01706-f005:**
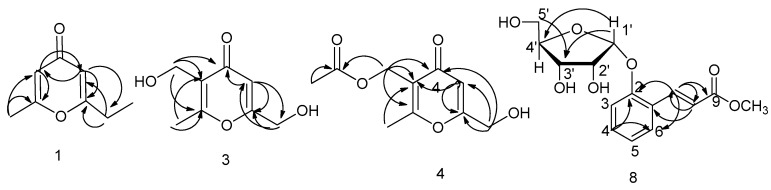
Key HMBC of Product 1, 3, 4 and 8.

**Figure 6 molecules-24-01706-f006:**
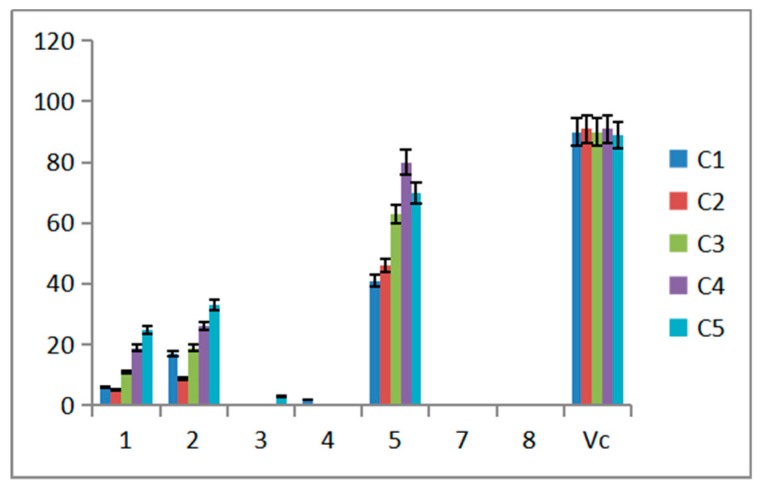
Scavenging effect on DPPH of compound.

**Table 1 molecules-24-01706-t001:** Chemical shifts of ^13^C-NMR in ribosome.

	C1	C2	C3	C4	C5
α-d-ribopyranose	94.3	70.8	71.1	681	63.3
β-d-ribopyranose	94.7	71.8	69.7	68.2	63.8
α-d-ribofuranose	97.1	71.7	70.8	83.8	63.3
β-d-ribofuranose	101.7	76.0	71.2	83.3	63.3

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
