# Peer review of "Secondary Metabolites and Antiradical Activity of Liquid Fermentation of Morchella sp. Isolated from Southwest China"

_molecules, 2019, doi:10.3390/molecules24091706_

Reviewer 1 Report

On the whole, the subject of the presented manuscript is quite interesting nevertheless need significant improvement. Language is poor and require native speaker improvement. Additionally, the methods, especially antioxidant, need enlargement. The presented studies are superficial. Personally, I believe that the sudies are interesting but expositive side is poor. Additionally, the following points should be improve:

Abstract: According to instructions for authors: The abstract should be a single paragraph and should follow the style of structured abstracts, but without headings

Morchella should be in italica

Results and discussion is superfiicial. These paragraphs need considerably amendment

Why Authors used only one antioxidant method? There are numerous valuable methods used for confirmation antioxidant activity

Why Authors used so high concentration of samples studied antioxidant activity?

References are insufficient   

Author Response

Comments and Suggestions for Authors

Abstract: According to instructions for authors: The abstract should be a single paragraph and should follow the style of structured abstracts, but without headings.

Response: Thank you for the referee’s kind advice. The abstract has been re-written without headings.

Morchella should be in italica

Response: Thanks for the referee’s suggestion. Morchella has been in italica

Results and discussion is superfiicial. These paragraphs need considerably amendment

Response: Thanks. The section has been re-written.

Why Authors used only one antioxidant method? There are numerous valuable methods used for confirmation antioxidant activity

Response: Thank you very much. The limitation of yield for the isolated compounds, only one method was selected to evaluate the antioxidant activity. So, the antioxidant activity has been changed into antiradical activity.

Why Authors used so high concentration of samples studied antioxidant activity?

Response: Thank you for the referee’s kind advice. The concentration of samples were consist with control VC.

References are insufficient   

Response: Thanks for the referee’s suggestion. References have been added.

Reviewer 2 Report

The manuscript of Yang et al. is interesting. Some comments and suggestions are below:

In the title please change antioxidant in antiradical because the only activity tested is based on the use of DPPH radical probe. Please take into account this consideration throughout the text.

Please modify Figure 6 introducing information about SEM or SD. Statistical analysis is full lacked.

In Materials and Methods section introduce the strumentation used for acquiring NMR and HR MS spectra. For HR MS data, please show error (ppm) between found and calculated mass for each compound.

Author Response

In the title please change antioxidant in antiradical because the only activity tested is based on the use of DPPH radical probe. Please take into account this consideration throughout the text.

Response: Thank you for the referee’s kind advice. The antioxidant activity has been changed into antiradical activity throughout the text.

Please modify Figure 6 introducing information about SEM or SD. Statistical analysis is full lacked. 

Response: Thanks for the referee’s suggestion. Statistical analysis have been added.

In Materials and Methods section introduce the strumentation used for acquiring NMR and HR MS spectra.

Response: Thanks for the referee’s suggestion. NMR spectra were acquired on aAgilent DD2400-MR spectrometer operating at 400 MHz (for proton NMR) and 100 MHz (for carbon NMR). High resolution mass spectra were obtained in the ABSciex TOF500+ MS using electrospray ionization

For HR MS data, please show error (ppm) between found and calculated mass for each compound. 

Response: Thank you for the referee’s kind advice. The calculated mass were added in text.

Reviewer 3 Report

I read with interest the manuscript entitled: “Secondary Metabolites and Antioxidant Activity of Liquid Fermentation of Morchella sp. Isolated from Southwest China”. I invested a considerable amount of time and effort reviewing and/or editing manuscripts. I think the authors have to be more careful when writing the manuscript. Thus, it is critical that authors address all the corrections and all queries. Discussions on both characterization of structures as well as on biological aspects need to be improved, these discussions are very poor. The manuscript provides a modest advance in the area.

Overall, the content of the manuscript needs to be improved because it has basic errors of elementary concepts. I believe that this manuscript could benefit of a major revision before publication in Molecules. I have attached a commented copy of the manuscript for the authors to consider in the revision.

Author Response

Response: Thanks for the referee’s suggestion and the commented copy of the manuscript. I’m sorry for the basic errors and have corrected the mistake throughout the text. 

Round  2

Reviewer 1 Report

Authors have improved the manuscript. Some points could be more precise nevertheless I can accept the present form. 

Reviewer 2 Report

The manuscript is clearly improved according to the reviewers' suggestions and comments.

A doubt is for the instrument ABSciex TOF500+ MS. Is it the real name? Please introduce city and state for instrumentations

Please check the SD bars. They seem similar each another. Introduce in the Materials and Methods section that data are expressed as mean value +/- SD